# Interventions to Mitigate Financial Toxicity in Adult Patients with Cancer in the United States: A Scoping Review

Seiichi Villalona [1,*], Brenda S. Castillo [1], Carlos Chavez Perez [1], Alana Ferreira [1], Isoris Nivar [2], Juan Cisneros [3] and Carmen E. Guerra [4,5,6]

1   Department of Medicine, Perelman School of Medicine, University of Pennsylvania, Philadelphia, PA 19104, USA; brenda.castillo@pennmedicine.upenn.edu (B.S.C.); carlos.chavezperez@pennmedicine.upenn.edu (C.C.P.); alana.ferreira@pennmedicine.upenn.edu (A.F.)
2   Division of Hematology/Oncology, Department of Medicine, Perelman School of Medicine, University of Pennsylvania, Philadelphia, PA 19104, USA; inivar@pennmedicine.upenn.edu
3   Institute for Systems Biology, Seattle, WA 98109, USA; cisneros.juand@gmail.com
4   Division of General Internal Medicine, Department of Medicine, Perelman School of Medicine, University of Pennsylvania, Philadelphia, PA 19104, USA; carmen.guerra@pennmedicine.upenn.edu
5   Abramson Cancer Center, University of Pennsylvania, Philadelphia, PA 19104, USA
6   Leonard Davis Institute of Health Economics, University of Pennsylvania, Philadelphia, PA 19104, USA
*   Correspondence: seiichi.villalona@pennmedicine.upenn.edu

**Abstract:** Financial toxicity adversely affects quality of life and treatment outcomes for patients with cancer. This scoping review examined interventions aimed at mitigating financial toxicity in adult patients with cancer and their effectiveness. We utilized five bibliographical databases to identify studies that met our inclusion criteria. The review included studies conducted among adult patients with cancer in the United States and published in English between January 2011 to March 2023. The review identified eight studies that met the inclusion criteria. Each of the studies discussed the implementation of interventions at the patient/provider and/or health system level. Collectively, the findings from this scoping review highlight both the limited number of published studies that are aimed at mitigating financial toxicity and the need to create and assess interventions that directly impact financial toxicity in demographically diverse populations of adult patients with cancer.

**Keywords:** financial toxicity; cancer care; intervention strategies

## 1. Introduction

Cancer is the second leading cause of death in the United States [1]. The American Cancer Society estimates that by 2023, 1.9 million new cancer cases will be diagnosed, and 609,820 cancer deaths will occur in the United States [2]. Although the advent of modern medical therapies has revolutionized cancer care with overall improved outcomes, such as survival rates, this advancement comes with a price that ultimately falls on the patients receiving care. This toll is felt physically, psychologically, and financially. On average, cancer treatments are four times more expensive than treatments for other common medical conditions [3]. Financial toxicity refers to the monetary burden of healthcare costs and associated consequences, and it encompasses the negative economic burden experienced by patients with cancer. Approximately 22–64% of patients with cancer report stress or worry about paying their medical bills [4]. For example, patients with cancer have a 2.5 increased likelihood of declaring bankruptcy relative to healthy individuals [5].

Financial toxicity in cancer treatments and its downstream effects are widespread, with several studies highlighting how higher degrees of financial toxicity have been associated with poorer health-related quality of life, medical adherence, survival, and treatment adherence [6–11]. As cancer care costs are projected to exceed USD 245 billion by 2030 in the United States alone [12], it is vital to address this widespread problem.

Despite the prevalence of financial toxicity in oncologic care, to the best of our knowledge, there is a limited number of published strategies and interventions to mitigate financial toxicity. Most of the published literature reviews about financial toxicity in oncology have focused on macro-level issues within the financial aspects of cancer care in the United States pertaining to healthcare policy [13], provided a general overview of financial toxicity in oncology [14], explored financial toxicity in a certain subset of patients with cancer [15,16], explored provider–patient communication regarding care costs [17], or studied financial toxicity with an international overview [18–20]. To our knowledge, one prior scoping review has examined interventions to address financial toxicity among adult patients with cancer [20]. This study aimed to identify interventions for reducing cancer-related financial toxicity and summarize their findings. The objective of this scoping review is to build on prior work and provide a comprehensive summary of the existing literature assessing interventions to mitigate financial toxicity in adult patients with cancer. This scoping review additionally was aimed at identifying the populations of adult patients with cancer largely excluded from the existent interventions. This scoping review aimed to identify interventions that have been studied among all types of patients with cancer, their effectiveness, and the barriers and/or facilitators of implementing these interventions.

## 2. Materials and Methods

### 2.1. Methodological Approach and Identifying Research Questions

The present scoping review was conducted using the frameworks of Arksey and O'Malley (2005) [21] and Levac, Colquhoun, and O'Brien (2010) [22]. The PRISMA Extension for Scoping Reviews (PRISMA-ScR) Checklist was additionally used to ensure appropriate domains were reported [23]. This scoping review aimed to assess interventions that mitigate financial toxicity in adult patients with cancer. The research questions guiding our search parameters included the following:

(1.) What published studies specifically assess or describe interventions to mitigate financial toxicity in adult patients with cancer?
(2.) Which interventions are and are not effective in adult patients with cancer?
(3.) Which populations of adult patients with cancer have been both included and excluded from these interventions?

### 2.2. Identification of Relevant Studies

The study team used five main bibliographic databases: PubMed, Psych Info, Scopus, CINAHL (Cumulative Index to Nursing and Allied Health Literature), and Econpapers. The Boolean search term combinations used in each of the databases included the following: "oncology" AND "financial toxicity"; "neoplasms" AND "financial toxicity"; "cancer" AND "financial toxicity"; "cancer" AND "financial hardship"; "cancer" AND "financial burden"; "cancer" AND "financial stress"; "cancer" AND "out-of-pocket"; "cancer" AND "economic burden"; "oncology" AND "financial toxicity" AND "financial hardship"; "oncology" AND "financial toxicity" AND "financial burden"; "oncology" AND "financial toxicity" AND "economic burden"; "oncology" AND "financial toxicity" AND "Cost-Benefit Analysis"; "oncology" AND "financial toxicity" AND "treatment cost"; "neoplasms" AND "financial toxicity" AND "financial hardship"; "neoplasms" AND "financial toxicity" AND "financial burden"; "neoplasms" AND "financial toxicity" AND "financial stress"; "neoplasms" AND "financial toxicity" AND "treatment cost"; "cancer" AND "financial toxicity" AND "financial hardship"; "cancer" AND "financial toxicity" AND "financial burden"; "cancer" AND "financial toxicity" AND "financial stress"; "cancer" AND "financial toxicity" AND "economic burden"; "cancer" AND "financial toxicity" AND "Cost-Benefit Analysis"; "cancer" AND "financial toxicity" AND "treatment cost".

### 2.3. Study Selection

We included peer-reviewed studies that met the following inclusion criteria: (1) studies on adult patients with cancer (i.e., 18+ years of age); (2) studies published in En-

glish; (3) studies conducted in the continental United States; (4) works published between 1 January 2011 and 31 March 2023. Studies with interventions for patients' family members, caretakers, and healthcare providers were included. The final studies included in the scoping review were limited to those evaluating, measuring the impact of, or reviewing interventions addressing financial toxicity in adult patients with cancer receiving treatment.

The decision to include studies in the United States was made to minimize confounding of the review findings when considering that the factors affecting financial toxicity (namely insurance coverage) vary significantly by country. Studies screening for financial toxicity and discussing financial toxicity without testing or describing an intervention were excluded. Studies were excluded if they were (1) basic science research conducted in laboratory settings on non-human subjects, (2) editorials or opinion pieces, (3) other literature reviews or data syntheses, (4) abstracts presented at scientific meetings, or (5) dissertations/theses.

The final list of search terms generated an initial set of articles imported into a Covidence library (*n* = 20,506) (Figure 1). The screening team (SV, BSC, and CCP) screened the abstracts (*n* = 3348) after the removal of duplicate studies (*n* = 17,158) (Figure 1). For the screening phase, the titles and abstracts of each article were independently assessed by two of the three screening team members for inclusion or exclusion. In total, 3304 studies were excluded in the screening phase with a Cohen's Kappa coefficient of 0.54, consistent with moderate interrater reliability. The remaining 41 articles were subjected to full-text review by the review team (SV, BSC, CCP, and AF) using a data extraction tool outside of the Covidence platform. Two of the four review team members independently assessed each study, and a third team member reconciled discordances in coding. This stage of the review process was iterative and involved multiple team meetings to discuss the remaining articles. The above methodology generated a final set of eight articles included for analysis. Figure 1 outlines the study selection methodology in the form of a PRISMA flow diagram.

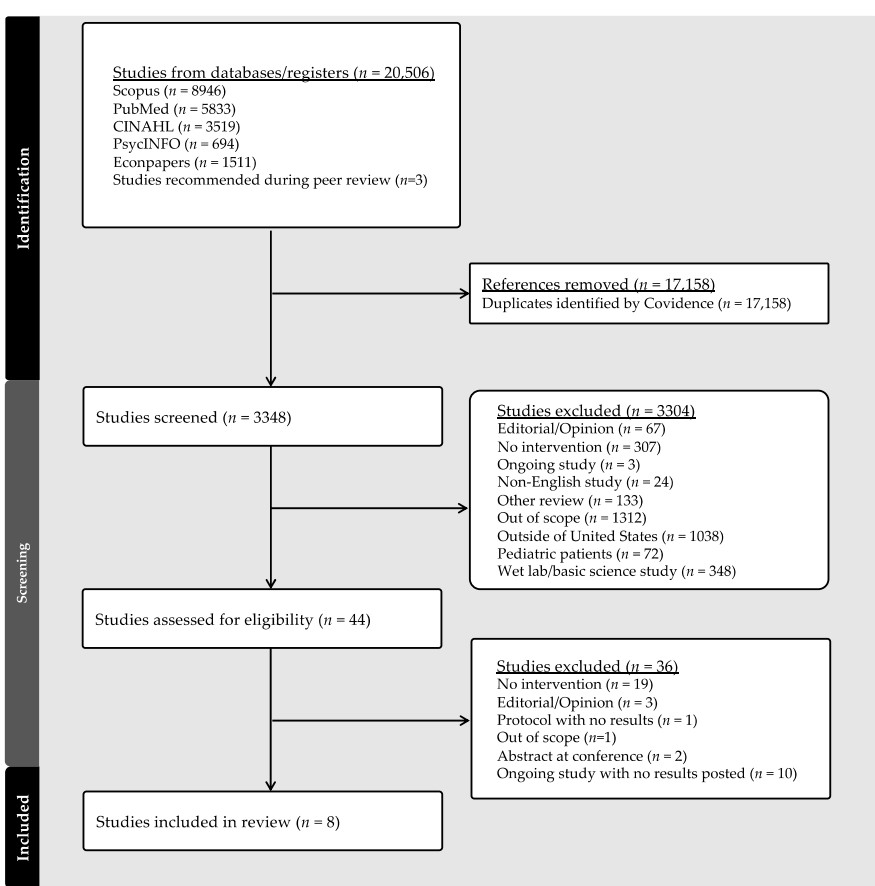

**Figure 1.** PRISMA flow diagram of article screening, review, and selection.

*2.4. Data Extraction*

Data were extracted and input into a study database created in Microsoft Excel (Microsoft Corp.). Variables of interest during this phase of the scoping review included the following: publication details (author, year of publication, region of the country); clinical specialty where intervention was implemented; intervention type and level of impact (patient, patient caretakers, medical providers, or health system); patient population by cancer diagnoses; race/ethnicities represented in these interventions; sex distribution of participants; study setting (urban or rural); and general summary of findings. The data extraction tool was piloted on three articles and modified during subsequent team meetings. Two of the four review team members (SV, BSC, CCP, and AF) read and reviewed each paper. A third team member not involved in the original review of any specific article reconciled discordances during team meetings. During this phase of the scoping review, the study team frequently convened to thoroughly discuss the trends and broader themes of the final articles.

## 3. Results

*3.1. Study Characteristics*

The iterative methodology employed in this scoping review generated a final set of eight articles to be included in the analyses. Table 1 outlines the findings of the studies included in this scoping review. The included studies were published after 2019. Among the studies including patients, the aggregate sample ($n$ = 697) was represented mainly by females ($n$ = 400, 57.4%), individuals diagnosed with solid malignancies ($n$ = 573, 82.2%), and those that identify as non-Hispanic White ($n$ = 495, 75.4%). Four studies were tailored explicitly to patient-level interventions; two were designed to include both a patient- and provider-level intervention, and two discussed interventions at the health system level. Most of the studies were conducted in outpatient clinical settings, whereby two were conducted by multidisciplinary departments, two were conducted explicitly by radiation oncology departments, and four were conducted by a medical oncology department. The studies included in the scoping review were conducted either in the Midwestern or Southern regions of the United States. Most of the studies were conducted in urban settings.

**Table 1.** Characteristics and summary of findings from studies assessing interventions mitigating financial toxicity in adult patients with cancer.

| Study Reference | Specialty | Cancer Types of Patients Included | Intervention Level | Intervention Description and Effectiveness | Patient Demographic Characteristics | Location |
|---|---|---|---|---|---|---|
| Kircher et al., (2019) [24] | Medical Oncology | Solid ($n$ = 95, 100%) | Patient-Level | Randomized controlled trial assessing the feasibility and acceptability of financial counseling. The intervention was considered both understandable and acceptable for the study participants. The intervention was not associated with significant decreases in financial distress. Higher emotional functioning and being married were associated with lower financial distress. | Mean age: 61.5 years Sex distribution - Males ($n$ = 43, 45.2%) - Females ($n$ = 52, 54.7%) Race/ethnicity distribution - Non-Hispanic White ($n$ = 67, 70.5%) - Non-Hispanic Black ($n$ = 20, 21.1%) - Other ($n$ = 8, 8.4%) Insurance coverage -Medicare/Medicaid ($n$ = 47, 49.5%) - Private insurance ($n$ = 48, 50.5%) | Chicago, Illinois; USA |
| Politi et al., (2020) [25] | Multidisciplinary | Solid ($n$ = 198, 96.1%) Hematologic ($n$ = 8, 3.9%) | Patient-Level | Personalized insurance decision aid ("I Can PIC") aimed at improving health insurance knowledge, decisional conflict, and decision self-efficacy. Successful in improving health insurance knowledge and confidence in understanding health insurance terms. No significant effect on financial toxicity. | Mean age: 52.7 years Sex distribution - Males ($n$ = 74, 35.9%) - Females ($n$ = 132, 64.1%) Race/ethnicity distribution - Non-Hispanic White ($n$ = 165, 80.1%) - Other ($n$ = 41, 19.9%) Insurance coverage - Employer-based ($n$ = 181, 87.9%) - Private insurance ($n$ = 22, 10.7%) - No insurance/self-pay ($n$ = 3, 1.4%) | Missouri and Illinois; USA |

**Table 1.** *Cont.*

| Study Reference | Specialty | Cancer Types of Patients Included | Intervention Level | Intervention Description and Effectiveness | Patient Demographic Characteristics | Location |
|---|---|---|---|---|---|---|
| Ning et al., (2020) [26] | Radiation Oncology | Solid ($n = 34$, 100%) | Health Systems-Level | Insurance coverage pilot that ensured preauthorization for proton beam therapy (PBT). Primary end points included patient enrollment, total cost of care with PBT use, and time to approval. Costs were compared between patients receiving PBT and patients receiving photon therapy. Average authorization time decreased from 17 days to <1 day ($p < 0.01$). Total overall medical costs did not demonstrate a significant difference between the groups. | Mean age: 62.5 years Sex distribution - Males ($n = 22$, 64.7%) - Females ($n = 12$, 35.3%) Race/ethnicity distribution - Not reported Insurance coverage - Not reported since insurance coverage was provided in intervention | Houston, Texas; USA |

**Table 1.** *Cont.*

| Study Reference | Specialty | Cancer Types of Patients Included | Intervention Level | Intervention Description and Effectiveness | Patient Demographic Characteristics | Location |
|---|---|---|---|---|---|---|
| Raghavan et al., (2021) [27] | Multidisciplinary | ----- | Health-System-Level | Financial Toxicity Tumor Board (FTTB). The FTTB was linked to a patient assistance program (PAP) for oncology pharmaceutical agents. The PAP served 3568 patients between 2019 and 2020. Personal expenditures saved totaled USD 50–60 million, and more than USD 1.3 million in copay assistance was provided for financially challenged patients. | ----- | Charlotte, North Carolina; USA |
| Tarnasky et al., (2021) [28] | Medical Oncology | Solid ($n = 195$, 95.5%) Hematologic ($n = 5$, 4.5%) | Patient-Level | Randomized controlled trial of a mobile health application intervention aimed at facilitating access to financial assistance programs for patients with cancer. This intervention was limited by completion of all aspects of the intervention, particularly the missing follow-up data. Participants in the intervention arm of the trial were more likely to apply for financial assistance programs. | Median age: 57 years Sex distribution - Males ($n = 92$, 46%) - Females ($n = 108$, 54%) Race/ethnicity distribution - Non-Hispanic White ($n = 141$, 70.5%) - Non-Hispanic Black ($n = 52$, 26%) - Others ($n = 7$, 3.5%) Insurance coverage - Medicare/Medicaid ($n = 46$, 23.5%) - Private insurance ($n = 137$, 67.0%) - Other ($n = 9$, 4.5%) | Durham, North Carolina; USA |

**Table 1.** *Cont.*

| Study Reference | Specialty | Cancer Types of Patients Included | Intervention Level | Intervention Description and Effectiveness | Patient Demographic Characteristics | Location |
|---|---|---|---|---|---|---|
| Hamel et al., (2022) [29] | Radiation Oncology | Solid ($n$ = 32, 100%) | Patient-Level Provider-Level | DIScussions of COst (DISCO) application. This application aimed to improve healthcare cost discussions between patients and providers. The intervention was associated with higher self-efficacy for managing treatment costs and facilitating patient–provider interactions/discussions. | Mean age: 61.5 years Sex distribution - Males ($n$ = 1, 3%) - Females ($n$ = 31, 97%) Race/ethnicity distribution - Non-Hispanic White ($n$ = 32, 100%) Insurance coverage - Medicare/Medicaid ($n$ = 17, 53%) - Employer-based ($n$ = 13, 41%) - Private insurance ($n$ = 2, 6%) | Detroit, Michigan; USA |
| Knight et al., (2022) [30] | Medical Oncology | Hematologic ($n$ = 107, 100%) | Patient-Level | A comprehensive intervention that utilized nurse navigators, clinical pharmacists, and community pro bono financial planners. Primary outcomes were improvement in mental and physical quality of life (QoL) and improvement in overall survival. The intervention had a higher QoL in physical and mental health scores ($p$ < 0.001). Lower mortality was observed in the patients who received the intervention relative to those who received standard of care. The intervention was associated with improved survival ($p$ = 0.03). | Median age: 58 years Sex distribution - Males ($n$ = 60, 56.1%) - Females ($n$ = 47, 43.9%) Race/ethnicity distribution - Non-Hispanic White ($n$ = 68, 63.6%) - Non-Hispanic Black ($n$ = 30, 28%) - Other (9, 8.4%) Insurance coverage - Medicare/Medicaid ($n$ = 61, 57%) - Private insurance ($n$ = 39, 36.5%) - Government insurance ($n$ = 3, 2.8%) - No insurance/self-pay ($n$ = 4, 3.7%) | Charlotte, North Carolina; USA |

**Table 1.** *Cont.*

| Study Reference | Specialty | Cancer Types of Patients Included | Intervention Level | Intervention Description and Effectiveness | Patient Demographic Characteristics | Location |
|---|---|---|---|---|---|---|
| Sadigh et al., (2022) [31] | Medical Oncology | Solid ($n$ = 19, 82.6%) Other ($n$ = 4, 17.4%) | Patient-Level Provider-Level | Assessment of the CostCOM intervention, which aimed at identifying out-of-pocket costs of patients newly diagnosed with cancer. The intervention included assessing patient–provider communication regarding out-of-pocket costs, counseling, and remote financial navigation. The intervention decreased patients' financial concerns ($p$ < 0.01), was acceptable, and was associated with high satisfaction among participants. | Median age: 61 years Sex distribution - Males ($n$ = 5, 21.7%) - Females ($n$ = 18, 78.3%) Race/ethnicity distribution - Non-Hispanic White ($n$ = 22, 95.6%) - Hispanic ($n$ = 2, 4.4%) Insurance coverage - Medicare/Medicaid ($n$ = 13, 56.5%) - Private insurance ($n$ = 8, 34.8%) - No insurance/self-pay ($n$ = 2, 8.7%) | Tennessee; USA |

*3.2. Summary of Findings*

3.2.1. Patient- and Provider-Level Interventions

The patient-level interventions included a novel application, a personalized health insurance decision aid, an individualized exercise program, and comprehensive patient assistance programs.

Kircher et al., (2019) [24] assessed the feasibility and acceptability of a financial counseling intervention. This study found that the intervention was understandable and acceptable for the participants in the trial. Although no significant decreases in financial distress were observed among the intervention group, the study found that patients with higher emotional functioning and those who were married were associated with lower financial distress.

Politi et al., (2020) [25] conducted a randomized controlled trial using a personalized insurance decision aid ("I Can PIC") aimed at improving health insurance knowledge, decisional conflict, and decision self-efficacy. Relative to controls, patients who received the intervention were found to have higher health insurance knowledge and confidence in understanding health insurance options. At a 3–6-month follow-up, no differences were observed in insurance knowledge, decisional conflict, decision self-efficacy, health insurance literacy, financial toxicity, or delayed care between patients in the intervention and control groups [25].

Tarnasky et al., (2021) [28] discuss the findings of a mobile health application-based trial aimed at educating about and facilitating access to financial assistance programs specifically for patients with cancer.

Hamel et al., (2022) [29] reported a pilot-tested DIScussions of COst (DISCO) application. This application aimed to improve healthcare cost discussions between patients and their oncology providers. The intervention was associated with higher self-efficacy for managing treatment costs and facilitating patient–provider interactions/discussions [29]. Similarly, Sadigh et al., (2022) [31] discuss the CostCOM intervention aimed at identifying out-of-pocket costs of patients newly diagnosed with cancer and facilitating patient–provider communication regarding these costs. The CostCOM intervention additionally included counseling and financial navigation.

Knight et al., (2022) [30] implemented a comprehensive patient assistance program. This program consisted of nursing-driven care navigation, financial counseling, social work, and pharmacy assistance. The Patient-Reported Outcomes Measurement Information System (PROMIS) and Comprehensive Score for Financial Toxicity (COST) tools were used to capture the patients' self-reported financial difficulties. Relative to the control group, patients who utilized the study intervention were associated with a significantly lower risk of death in multivariate analyses (adjusted hazard ratio (aHR) 0.44, 95% confidence interval 0.21–0.94; $p = 0.03$) [30].

3.2.2. Health-System-Level Interventions

Interventions implemented at the health system level included a pilot program for insurance coverage for patients receiving radiation therapy and the creation of a Financial Toxicity Tumor Board (FTTB) with an associated patient assistance program.

Ning et al., (2020) [26] discuss the piloting of an insurance coverage intervention, specifically aiming to increase access to proton therapy. The study found that the intervention significantly increased patient access to this therapeutic modality by decreasing the time of prior authorization for proton therapy from 17 days to <1 day ($p < 0.01$) [26]. The total overall medical costs for patients did not significantly differ between the control and treatment groups during the 3-year study follow-up period [26].

The FTTB discussed by Raghavan et al., (2021) [27] involved patients being referred to the multidisciplinary team after being screened for financial toxicity using an electronic patient assessment tool. This intervention helped address issues including insurance coverage, copay assistance, payer impediments, and coding and billing problems.

This intervention was reported to be associated with a USD 55–60 million reduction in oncology care costs over two years [27].

## 4. Discussion

The findings from this scoping review summarize the limited studies that assess interventions to help mitigate financial toxicity among adult patients with cancer. The recent emerging literature on this topic has included interventions at the patient and health system levels. Most studies included in this scoping review discussed patient-centered interventions [24,25,28–31]. Most of the interventions were designed to directly address different aspects of financial toxicity, including capacity building [29], knowledge acquisition, insurance or healthcare costs [25,28,29,31], and immediate cost reduction for patients receiving oncology care (e.g., copay assistance, medication cost coverage or discounts) [24,27,28,30]. Two of the studies assessed the financial benefit at the health system level versus the immediate or long-term cost to the patient receiving care [26,27].

Several noteworthy characteristics emerged among the studies included in this review. First, the articles that met the inclusion criteria in this scoping review highlight the underrepresentation of patients from non-White racial/ethnic groups. Previous studies have identified higher financial toxicity from receiving cancer care among patients from underrepresented racial and ethnic groups relative to their non-Hispanic White counterparts [32–34]. This finding points to the need to design interventions aimed at mitigating financial toxicity among established at-risk groups of patients with cancer. Second, most of the population from the studies included in this scoping review was represented by patients with cancer who self-identify as females. Prior studies have reported female sex as a demographic group of patients with cancer at higher risk of experiencing financial toxicity [35,36]. The relative underrepresentation of males in interventional studies calls for a better understanding of financial toxicity among patients with cancer who identify as male, as well as the increasing overall participation in interventions among this demographic group. Third, the studies included in this scoping review were conducted across urban areas in southern or midwestern states. Longer distance from cancer treatment centers has been identified as one of the patient characteristics associated with experiencing financial toxicity [37]. The paucity of studies aimed at addressing some of the physical/geographic aspects of receiving specialized cancer care further highlights the need for interventions that mitigate these known structural barriers for this patient population. Lastly, the studies included in this review are from interventions implemented among primarily patients with solid malignancies. This underpins the importance of conducting future studies addressing financial toxicity among patients with hematologic malignancies.

*Topically Related Non-Intervention Studies of Financial Toxicity in Oncology*

This scoping review aimed to assess interventions to mitigate financial toxicity among adult patients with cancer. Studies designed to screen or effectively identify patients at risk for financial toxicity were excluded from the final analyses. However, we identified studies that aimed to stratify at-risk patients receiving oncology care and properly implement interventions to address financial toxicity.

Among patient-level studies, several projects have aimed to produce patient-reported measures that identify individuals at risk of significant financial burden. One such measure is the COST tool, an 11-item assessment questionnaire. De Souza et al., (2017) [38] validated the test reliability and internal consistency of the COST tool. They examined its relationship with patient-derived data, including sociodemographic information, psychological distress, emergency room visits, inpatient admissions, willingness to discuss healthcare costs, and quality of life, via multivariable analyses. This study found that lower income, psychological distress, unemployment status, and higher number of inpatient admissions were significantly correlated with lower COST scores which signify increased financial toxicity [38].

Subsequent studies have further developed the COST tool to be utilized within different clinical contexts in cancer care. D'Rummo et al., (2018) [39] used the COST-Functional Assessment of Chronic Illness Therapy (COST-FACIT) scoring system in radiation oncology. This study included 167 patients and showed that this intervention was successful in identifying financial toxicity in the radiation oncology setting while additionally reporting high toxicity to be associated with non-married individuals and patients under 65 years of age [39]. The COST measure was subsequently incorporated into the Lessening the Impact of Financial Toxicity (LIFT) program, which utilizes COST scores to identify patients at high risk of financial toxicity and connects them with oncology financial navigators as described by Wheeler and colleagues (2022) [40]. The program was associated with a 7-point COST score improvement in patient-reported financial distress [40]. Wheeler et al., (2022) [40] identified several core components that are necessary for the implementation and effectiveness of LIFT, including the systematic cataloging of information on a patient's eligibility for financial navigation, the development of an ongoing relationship between patients and financial navigators, willingness to engage with and implement financial navigation, trained implementation of LIFT by navigators, and an existent case management system [40].

Other measures for screening for financial toxicity have also been designed with positive results. Prasad and colleagues (2021) [41] illustrated a distinct screening tool for patients receiving radiation therapy that uses a comprehensive survey to identify patient-reported factors linked to the loss of income, job, spouse, or difficulty paying for meals [41]. The tool uses three patient-derived variables to calculate risk: age, money owed, and worries about copay. After analysis using a logistic regression model, it was determined that 34 (22%) out of 157 respondents were experiencing financial toxicity [41]. Identifying individuals at risk for financial toxicity remains critical for delivering toxicity-mitigating interventions and improving health and socioeconomic outcomes.

At the health system level, some researchers have examined factors outside of the individual characteristics of patients with cancer and the clinical environment where they receive care as there are other aspects of financial toxicity that could limit cancer care. Khorsandi and Giancola et al., (2023) [42] sought to create a "Housing in Cancer" workgroup of distinct government, non-profit, and medical agencies to formulate interventions for housing insecurity related to cancer in the New Orleans region [42]. The authors discuss that the health systems harboring the patient are composed of weakly connected stakeholder groups that lack financial motivation and divert responsibility [42]. They concluded that the terminology for who is "at-risk" for housing insecurity is distinct between health care and housing services (using the poverty line and the area median income, respectively); thus, a standard shared definition may improve network collaboration [42]. The conclusions yielded from the workgroup analysis point to the importance of cross-sector collaborative efforts to address large-scale financial toxicity through ingenuity and partnership.

Furthermore, direct consideration of patients' perspectives is imperative when designing interventions to offer the best support. Shankaran et al., (2017) [43] interviewed twenty-one patients with colorectal or breast cancer on the impact of cancer on their finances, employment status, and their opinions on developing a novel financial literacy course [43]. Seventy-six percent of respondents declared that a financial literacy course that focuses on navigating the financial burden of cancer and identifying assistance resources would benefit them [43].

## 5. Conclusions

Despite the rigorous methodology, the present scoping review has some noteworthy limitations. First, including only English-language articles and studies conducted within the United States may lead to the exclusion of relevant studies from other countries or those published in different languages. Second, our focus on the United States healthcare system might limit the generalizability of our findings to other healthcare settings with different levels of insurance coverage and financial structures. The decision to exclude

these works minimizes the heterogeneity of studies in different cultural and health system contexts. Similarly, pediatric studies were excluded to facilitate data analysis as pediatric cancer populations have distinct socioeconomic factors and vary in interventions that may impact financial toxicity.

This scoping review has several strengths. To our knowledge, this is the first published scoping review examining interventions aimed at addressing financial toxicity in adult patients with cancer in the United States that also discusses which populations have largely been excluded from these initiatives. Additionally, this study was not limited to patients with a specific group of malignancies. The methodological approach undertaken in this scoping review was multifaceted and iterative, minimizing the potential for bias in the findings or conclusions.

The limited and recently emerging interventions focused on mitigating financial toxicity are heterogeneous regarding the intervention modalities and level of impact. This poses a challenge for researchers and clinicians to best determine the most efficacious strategies for mitigating financial toxicity among patients with cancer at all levels of the care continuum. The existing literature has primarily included females, patients with solid malignancies, and those identifying as non-Hispanic White and has only been implemented in a few geographic regions in the United States. Future studies should particularly focus on examining how financial toxicity is experienced among the demographic groups largely excluded from the current literature. Collectively, these findings call for interventions directly impacting financial toxicity in demographically and geographically diverse populations of adult patients with cancer receiving care in the United States.

**Author Contributions:** Conceptualization, A.F., I.N., J.C. and C.E.G.; methodology, S.V., B.S.C., C.C.P. and A.F.; analytic software management, S.V.; formal analysis, S.V., B.S.C., C.C.P. and A.F.; data curation, S.V., B.S.C., C.C.P. and A.F.; writing—original draft preparation, S.V., B.S.C., C.C.P. and A.F.; writing—review and editing, S.V., B.S.C., C.C.P., A.F., I.N., J.C. and C.E.G.; supervision and project administration, S.V. All authors have read and agreed to the published version of the manuscript.

**Funding:** This research received no external funding.

**Informed Consent Statement:** Not applicable.

**Acknowledgments:** The authors gratefully acknowledge Maylene Qiu, Systematic Review Coordinator and Clinical Liaison Librarian at the University of Pennsylvania, for her assistance with this review.

**Conflicts of Interest:** C.E.G. was supported by NCI grant P30 CA 016520. The NCI did not play a role in the review process.

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
