# Peer review of "Interventions to Mitigate Financial Toxicity in Adult Patients with Cancer in the United States: A Scoping Review"

_curroncol, doi:10.3390/curroncol31020068_

Round 1

Reviewer 1 Report (Previous Reviewer 1)

Comments and Suggestions for Authors

Thank you for re-submitting this article for review. The authors have made significant improvements to the manuscript and the inclusion of additional articles with a broader scope makes it more widely applicable.

However, I still have some concerns that require addressing by the authors. I would encourage the authors to re-visit their search terms and inclusion/exclusion criteria as I believe there are possibly some studies missing which appear to meet criteria (see below). Additionally, some of the new studies included appear to be outside the scope of the criteria.

Apologies, there was an error in my previous review the similar review was published in Critical Review in Oncology and Haematology - Interventions for financial toxicity among cancer survivors: A scoping review - PubMed (nih.gov)

Specific Comments

Introduction

·       There has been a recent scoping review published on interventions for financial toxicity - Interventions for financial toxicity among cancer survivors: A scoping review - PubMed (nih.gov). Whilst this was not US specific almost all of the articles are from the US

Methods

·       The inclusion and exclusion criteria should specifically state what was defined as an intervention to address financial toxicity. Was this any study that looked at reducing out of pocket expenses for patients?

Results

·       Ning et al – It does not appear that this manuscript was designed as an intervention to reduce financial toxicity for patients and the authors should review whether this meets inclusion criteria

·       Wonders et al. Similar to above, this was an exercise program that was not specifically designed to reduce levels of financial toxicity.

·       3.2.2 – The sections discussing the studies by Ning et al and Raghavan et al. focus on cost of care overall and not patient costs or financial toxicity

Discussion

·       Section 4.1 is a good addition to the manuscript, although is very wordy and should be condensed to allow for a more succinct discussion. The paragraph describing the article by Khorsandi and Giancola could be reduced to a few sentences.

·       There is no mention in the discussion about the limited efficacy of the included interventions. A paragraph discussing this and some recommendations for future directions and research would be warranted. Especially as one of the research questions is to determine which interventions are effective in reducing financial toxicity

     Conclusion - I think that it also needs to be made clearer that there is limited evidence to guide the most efficacious strategy to reduce financial toxicity in patients with cancer

Studies not included that could possibly meet inclusion criteria. I think that the inclusion and exclusion criteria need to be more clearly defined to improve the reproducibility of the study

·       Sadigh et al 2022 - Treatment out-of-pocket cost communication and remote financial navigation in patients with cancer: a feasibility study - PubMed (nih.gov)

·       Kircher et al. 2019 - Piloting a Financial Counseling Intervention for Patients With Cancer Receiving Chemotherapy - PubMed (nih.gov)

·       Tarnasky et al. 2021 - https://pubmed.ncbi.nlm.nih.gov/33797952/

Comments on the Quality of English Language

Overall well written article

Author Response

Thank you for re-submitting this article for review. The authors have made significant improvements to the manuscript and the inclusion of additional articles with a broader scope makes it more widely applicable.

However, I still have some concerns that require addressing by the authors. I would encourage the authors to re-visit their search terms and inclusion/exclusion criteria as I believe there are possibly some studies missing which appear to meet criteria (see below). Additionally, some of the new studies included appear to be outside the scope of the criteria.

Apologies, there was an error in my previous review the similar review was published in Critical Review in Oncology and Haematology - Interventions for financial toxicity among cancer survivors: A scoping review - PubMed (nih.gov)

Specific Comments

Introduction

  • There has been a recent scoping review published on interventions for financial toxicity - Interventions for financial toxicity among cancer survivors: A scoping review - PubMed (nih.gov). Whilst this was not US specific almost all of the articles are from the US

-Response: Thank you for your thoughtful feedback. We included the review mentioned in the new version of the introduction. What our scoping review highlights is a more qualitative analysis of the populations of patients with cancer included in the existent studies. Our manuscript helps highlight how particular groups (males, ethnic/racial minorities, patients with hematologic malignancies, and certain parts of the US) have been excluded from the published interventions.

Methods

  • The inclusion and exclusion criteria should specifically state what was defined as an intervention to address financial toxicity. Was this any study that looked at reducing out of pocket expenses for patients?

-Response: Thank you for your thoughtful feedback. Out-of-pocket expenses was not specifically included as a search term in our search schema for the review. While we recognize that this could be a methodological limitation, we don’t believe that it hinders our overall findings. We can see where this would be a limitation if we were conducting a systematic review. We believe that the methodological approach of our scoping review yielded noteworthy findings that helps advance where future work in financial toxicity should explore further.

Results

  • Ning et al – It does not appear that this manuscript was designed as an intervention to reduce financial toxicity for patients and the authors should review whether this meets inclusion criteria
  • Wonders et al. Similar to above, this was an exercise program that was not specifically designed to reduce levels of financial toxicity.
  • 3.2.2 – The sections discussing the studies by Ning et al and Raghavan et al. focus on cost of care overall and not patient costs or financial toxicity

Response: Thank you for your thoughtful feedback. Because of the limited body of literature we kept these studies in the final set because they speak to interventions aimed at reducing financial toxicity at the health systems level that also have an impact at the patient level (namely longer hospital stays and insurance coverage of treatment).

Discussion

  • Section 4.1 is a good addition to the manuscript, although is very wordy and should be condensed to allow for a more succinct discussion. The paragraph describing the article by Khorsandi and Giancola could be reduced to a few sentences.
  • There is no mention in the discussion about the limited efficacy of the included interventions. A paragraph discussing this and some recommendations for future directions and research would be warranted. Especially as one of the research questions is to determine which interventions are effective in reducing financial toxicity

     Conclusion - I think that it also needs to be made clearer that there is limited evidence to guide the most efficacious strategy to reduce financial toxicity in patients with cancer

 Studies not included that could possibly meet inclusion criteria. I think that the inclusion and exclusion criteria need to be more clearly defined to improve the reproducibility of the study

  • Sadigh et al 2022 - Treatment out-of-pocket cost communication and remote financial navigation in patients with cancer: a feasibility study - PubMed (nih.gov)
  • Kircher et al. 2019 - Piloting a Financial Counseling Intervention for Patients With Cancer Receiving Chemotherapy - PubMed (nih.gov)
  • Tarnasky et al. 2021 - https://pubmed.ncbi.nlm.nih.gov/33797952/

Response: Thank you for your thoughtful feedback. We have included all the recommendations from the reviewer in this version of the manuscript. We are very appreciative of the reviewer’s comments in helping improve out manuscript.

Reviewer 2 Report (New Reviewer)

Comments and Suggestions for Authors

This paper is a review of publications and presentations related to financial toxicity which is a significant issue for patients with cancer and their families/- the authors ultimately discuss 6 publications in this are after a detailed review and analysis of 5 online search tools such as pubmed.

the results are presented in a clear table and the discussion highlights the knowledge gaps that persist including minority and non -urban population representation . The studies reviewed in the analysis are limited to the continental United States. I was disappointed to see that only 6 studies were included despite a large number of articles being assessed by the group particularly as this issue is a significant one for patients and families.

The report could be improved by :

changing the title to indicate that the study only reflects US studies

a figure ( to compliment the discussion ) highlighting current solutions on the right side of a patient and potential future solutions on the left including studies of minorities and rural populations- this would help in disseminating the study on social media 

Author Response

This paper is a review of publications and presentations related to financial toxicity which is a significant issue for patients with cancer and their families/- the authors ultimately discuss 6 publications in this are after a detailed review and analysis of 5 online search tools such as pubmed. The results are presented in a clear table and the discussion highlights the knowledge gaps that persist including minority and non -urban population representation . The studies reviewed in the analysis are limited to the continental United States. I was disappointed to see that only 6 studies were included despite a large number of articles being assessed by the group particularly as this issue is a significant one for patients and families.

The report could be improved by :changing the title to indicate that the study only reflects US studies

-Response: Thank you for this feedback. We agree with the reviewer and have made these changes to the title of the manuscript.

A figure ( to compliment the discussion ) highlighting current solutions on the right side of a patient and potential future solutions on the left including studies of minorities and rural populations- this would help in disseminating the study on social media 

-Response: Thank you for the recommendation. We can provide a separate figure to help with dissemination should the article be accepted for publication.

Round 2

Reviewer 1 Report (Previous Reviewer 1)

Comments and Suggestions for Authors

Thank you for addressing the previous concerns that were raised

Comments on the Quality of English Language

Well written

Author Response

Thank you for the thoughtful feedback in improving our manuscript throughout the peer review process.

Reviewer 2 Report (New Reviewer)

Comments and Suggestions for Authors

The comments raised in my initial review have been addressed - I note that the authors will provide a figure if the article is accepted for publication 

Author Response

Thank you for the thoughtful feedback in improving our manuscript throughout the peer review process.

This manuscript is a resubmission of an earlier submission. The following is a list of the peer review reports and author responses from that submission.

Round 1

Reviewer 1 Report

Comments and Suggestions for Authors

Thank you for submitting a review on this important and under-researched topic.

The authors have submitted a scoping review exploring interventions in the USA to address financial toxicity in cancer survivors. Whilst mitigation of financial toxicity amongst cancer survivors is an important topic, I have some major concerns about the review.

1. The review has a very narrow focus which limits its applicability and generalizability to a broader audience. A scoping review of 2 articles makes it very challenging to draw any conclusions. I would suggest if wanting to limit to the USA and interventional studies only then discussion of trials in progress would also be essential to be able to allow a more robust discussion. For example, this study protocol - https://trialsjournal.biomedcentral.com/articles/10.1186/s13063-022-06344-3 

2. Although I agree with the points that have been raised in the discussion regarding which groups of patients are more at risk of financial toxicity, drawing these conclusions from the 2 papers is very challenging - and there are another many factors which could be discussed. It is not clear how these factors were decided upon as the major issues.

3. There has been a more comprehensive review published on the same topic very recently in Supportive Care in Cancer

4. Important components of financial toxicity interventions such as financial counselling and navigation are not discussion in detail. There is no mention of these in the discussion.

Comments on the Quality of English Language

Overall, I have no concerns about the quality of the English

Author Response

Thank you for submitting a review on this important and under-researched topic.

The authors have submitted a scoping review exploring interventions in the USA to address financial toxicity in cancer survivors. Whilst mitigation of financial toxicity amongst cancer survivors is an important topic, I have some major concerns about the review.

  1. The review has a very narrow focus which limits its applicability and generalizability to a broader audience. A scoping review of 2 articles makes it very challenging to draw any conclusions. I would suggest if wanting to limit to the USA and interventional studies only then discussion of trials in progress would also be essential to be able to allow a more robust discussion. For example, this study protocol - https://trialsjournal.biomedcentral.com/articles/10.1186/s13063-022-06344-3 

-Response: Thank you for the thoughtful feedback. We agree with the reviewer’s assessment regarding the difficulty in generating conclusions from a limited number of studies. This scholarly topic is very under-researched, which further points to the need of additional studies that examine financial toxicity. Taking this into consideration, we also acknowledge that our definition of interventions may have been too strict. For this reason, we liberalized our definitions and did not limit our final inclusion in the analytic sample of articles to those with a control group. While we agree that including some studies that are actively being conducted would help expand the discussion, our team felt it would be difficult to systematically capture all the ongoing studies. We included an additional section to the “Discussion” section that further expands on topically related studies to financial toxicity that were not specifically an intervention.

  1. Although I agree with the points that have been raised in the discussion regarding which groups of patients are more at risk of financial toxicity, drawing these conclusions from the 2 papers is very challenging - and there are another many factors which could be discussed. It is not clear how these factors were decided upon as the major issues.

-Response: Thank you for the thoughtful feedback. Please see response above.

  1. There has been a more comprehensive review published on the same topic very recently in Supportive Care in Cancer

-Response: Thank you for the feedback. Upon assessment of the recently published review articles in Supportive Care in Cancer, it appears that the reviews published between 2022-2023 are either focused on either patients with a particular groups of cancers (e.g. breast) or in other countries. Our review primarily focused on the US to limit variability across the different models of healthcare coverage across countries that would how financial toxicity would impact patients with cancer. Please let us know if there was a different review that you were referring to.

  • Çeli̇k, Yusuf, et al. "Evaluation of financial toxicity and associated factors in female patients with breast cancer: a systematic review and meta-analysis." Supportive Care in Cancer12 (2023): 691.
  • Bian, Jingru, et al. "Financial toxicity experienced by patients with breast cancer-related lymphedema: a systematic review." Supportive Care in Cancer6 (2023): 354.
  • Kuang, Yi, et al. "Communication of costs and financial burdens between cancer patients and healthcare providers: a qualitative systematic review and meta-synthesis." Supportive Care in Cancer3 (2023): 192.
  • Donkor, Andrew, et al. "Financial toxicity of cancer care in low-and middle-income countries: a systematic review and meta-analysis." Supportive Care in Cancer9 (2022): 7159-7190.
  • Udayakumar, Suji, et al. "Cancer treatment-related financial toxicity experienced by patients in low-and middle-income countries: a scoping review." Supportive Care in Cancer8 (2022): 6463-6471.

The scoping reviews listed above were included in the new introduction section of the manuscript.

  1. Important components of financial toxicity interventions such as financial counselling and navigation are not discussion in detail. There is no mention of these in the discussion.

-Response: Thank you for the feedback. Upon broadening our criteria, we were able to include studies that address this.

Reviewer 2 Report

Comments and Suggestions for Authors

This paper addresses an important and clearly under-studied area in health policy, the financial toxicity associated with high-cost care, specifically cancer care. The authors provide meaningful justification for the need to focus on how to address the impacts of extremely high costs of care when patients are diagnosed with cancer. 

Unfortunately, the focus of the paper seems to be misplaced and the articles that were included in the study do not really appear to address interventions designed to impact financial toxicity. First, the finding that only two articles out of an initial more than 17,000 articles met the criteria for inclusion in the literature review suggests that the topic of the review might be too focused and therefore result in findings that are not broadly generalizable or useful to the audience. In this case, the focus on actual interventions implemented to address financial toxicity led to the exclusion of a number of articles that might have provided additional insights into the causes of financial toxicity in cancer care and, by extension, led to potential solutions or additional interventions designed to address this problem. Since so few interventions have actually been attempted, a more compelling article might have refocused the literature search and abstraction to broaden the topic and provide more of a complete picture of financial toxicity in cancer care.

Second, the two articles included in the paper do not really involve interventions designed to impact financial toxicity, but instead appear to involve interventions designed to help patients and caregivers better understand the costs they will incur now that the patient has been diagnosed with cancer. From what I understand, the costs will still be incurred, but the patients perhaps understand them better than they otherwise would have in the absence of the intervention. From my perspective, a true intervention designed to reduce financial toxicity would encompass patient and provider decision-making, outreach to additional potential sources of financial assistance, and additional supports related to mitigating the economic impacts of cancer treatment. Neither of these studies really go that far.

If this paper is submitted again, I would also recommend the authors carefully read it to catch any errors. For example, in section 2.4 on Data Extraction, the authors note that 2 authors did a given task, but list four sets of initials in parentheses. Typos and other errors of this nature appear throughout the paper and raise concerns about the overall quality of the analysis. As one other example, why did the authors list "references from other sources (n=0)" in Figure 1, when it seems clear they simply did not look for references from other sources?

Comments on the Quality of English Language

The quality of the English language used in this draft paper is moderate. If the paper is eventually accepted for publication, I would recommend review by a copy-editor to address grammatical and other errors that appear throughout the manuscript. But these are mostly minor (with the exceptions noted above in my main comments).

Author Response

This paper addresses an important and clearly under-studied area in health policy, the financial toxicity associated with high-cost care, specifically cancer care. The authors provide meaningful justification for the need to focus on how to address the impacts of extremely high costs of care when patients are diagnosed with cancer. 

Unfortunately, the focus of the paper seems to be misplaced and the articles that were included in the study do not really appear to address interventions designed to impact financial toxicity. First, the finding that only two articles out of an initial more than 17,000 articles met the criteria for inclusion in the literature review suggests that the topic of the review might be too focused and therefore result in findings that are not broadly generalizable or useful to the audience. In this case, the focus on actual interventions implemented to address financial toxicity led to the exclusion of a number of articles that might have provided additional insights into the causes of financial toxicity in cancer care and, by extension, led to potential solutions or additional interventions designed to address this problem. Since so few interventions have actually been attempted, a more compelling article might have refocused the literature search and abstraction to broaden the topic and provide more of a complete picture of financial toxicity in cancer care.

Second, the two articles included in the paper do not really involve interventions designed to impact financial toxicity, but instead appear to involve interventions designed to help patients and caregivers better understand the costs they will incur now that the patient has been diagnosed with cancer. From what I understand, the costs will still be incurred, but the patients perhaps understand them better than they otherwise would have in the absence of the intervention. From my perspective, a true intervention designed to reduce financial toxicity would encompass patient and provider decision-making, outreach to additional potential sources of financial assistance, and additional supports related to mitigating the economic impacts of cancer treatment. Neither of these studies really go that far.

-Response: Thank you for the thoughtful feedback. We agree with the reviewer’s overall assessment and have addressed these concerns. Please see responses to Reviewer 1 for further clarification. We broadened our criteria and included additional studies into the final analytic sample.

If this paper is submitted again, I would also recommend the authors carefully read it to catch any errors. For example, in section 2.4 on Data Extraction, the authors note that 2 authors did a given task, but list four sets of initials in parentheses. Typos and other errors of this nature appear throughout the paper and raise concerns about the overall quality of the analysis. As one other example, why did the authors list "references from other sources (n=0)" in Figure 1, when it seems clear they simply did not look for references from other sources?

-Response: Thank you for the thoughtful feedback. We attempted to further clarify how the study team approached the screening and review of articles. We originally included the “references from other sources” in Figure 1 as it was part of the PRISMA flow diagram. We have removed this section to make it clearer.

Reviewer 3 Report

Comments and Suggestions for Authors

The manuscript presents a systematic review on interventions designed to mitigate the financial toxicity caused by cancer treatment among adult patients in the United States. The subject is very interesting in the context of oncology, and the text is generally well-written.

There are minor changes required in the manuscript, described below:

(1) In general, papers include the objective of the study within the Introduction section. If it is not a section required by the journal, it would be better to exclude the Objective section, and include the sentence describing the objective in the previous section;

(2) There is no need to include the definition and the importance of scoping reviews in the beginning of the Materials and Methods section, therefore, it would be better to exclude the two first sentences of the subsection "2.1 Methodological Approach and Identifying Research Questions" (lines 80-82, page 2);

(3) The focus of the scoping review (lines 86-87, page 3) should include the geographical delimitation to studies performed with patients in the United States;

(4) The phrasing of the first research question that guided the literature search (lines 89-90, page 3) seems oddly informal in relation to the text, please, consider rephrasing;

(5) The sentence defining the decision to limit the search to continental United States seem a little bit confusing (lines 122-123, page 3), suggestion to rephrase it to: "when considering that the factors affecting financial toxicity (namely insurance coverage) vary significantly by country";

(6) The definition of the study team involved in the development of the literature review varies along the description of Materials and Methods section (e.g., three individuals in line 129, page 3; four individuals in lines 135 and 149-150, page 4 - although the sentence in lines 149-150 indicates "Two team members"). Therefore, it would be better to make an uniform definition of the study team (i.e., researchers involved in the study - should correspond to the authors of the paper), and the team involved in each stage of literature screening (literature search, screening, abstract and full-text review);

(7) Check the text for typos (e.g., "im-prove" in Table 1, column "Intervention description and effectiveness");

(8) It is necessary to revise page numbering, since the inclusion of the table in landscape format seemed to have distorted the page count.

Author Response

The manuscript presents a systematic review on interventions designed to mitigate the financial toxicity caused by cancer treatment among adult patients in the United States. The subject is very interesting in the context of oncology, and the text is generally well-written.

There are minor changes required in the manuscript, described below:

(1) In general, papers include the objective of the study within the Introduction section. If it is not a section required by the journal, it would be better to exclude the Objective section, and include the sentence describing the objective in the previous section;

-Response: Thank you for the feedback. These changes have been reflected in the current version of the manuscript.

(2) There is no need to include the definition and the importance of scoping reviews in the beginning of the Materials and Methods section, therefore, it would be better to exclude the two first sentences of the subsection "2.1 Methodological Approach and Identifying Research Questions" (lines 80-82, page 2);

-Response: Thank you for the feedback. These changes have been reflected in the current version of the manuscript.

(3) The focus of the scoping review (lines 86-87, page 3) should include the geographical delimitation to studies performed with patients in the United States;

-Response: Thank you for the feedback. We have included this language in the “2.3 Study Selection” section of the manuscript

(4) The phrasing of the first research question that guided the literature search (lines 89-90, page 3) seems oddly informal in relation to the text, please, consider rephrasing;

-Response: Thank you for the feedback. We have included rephrased the first research question in the “2.1 Methodological Approach and Identifying Research Questions” Section of the manuscript.

(5) The sentence defining the decision to limit the search to continental United States seem a little bit confusing (lines 122-123, page 3), suggestion to rephrase it to: "when considering that the factors affecting financial toxicity (namely insurance coverage) vary significantly by country";

-Response: Thank you for the feedback. We have modified the sentences per the reviewer’s suggestion in the “2.3 Study Selection” Section of the manuscript.

(6) The definition of the study team involved in the development of the literature review varies along the description of Materials and Methods section (e.g., three individuals in line 129, page 3; four individuals in lines 135 and 149-150, page 4 - although the sentence in lines 149-150 indicates "Two team members"). Therefore, it would be better to make an uniform definition of the study team (i.e., researchers involved in the study - should correspond to the authors of the paper), and the team involved in each stage of literature screening (literature search, screening, abstract and full-text review);

-Response: Thank you for the feedback. We attempted to clarify how the study team approached the screening and reconciliation of articles included in the study.

(7) Check the text for typos (e.g., "im-prove" in Table 1, column "Intervention description and effectiveness");

-Response: Thank you for the feedback. We reformatted the manuscript template provided by Current Oncology to address this.

(8) It is necessary to revise page numbering, since the inclusion of the table in landscape format seemed to have distorted the page count.

-Response: Thank you for the feedback. We have reformatted the page numbers.